# Evidence from the first Shared Medical Appointments (SMAs) randomised controlled trial in India: SMAs increase the satisfaction, knowledge, and medication compliance of patients with glaucoma

**Nazlı Sönmez**[1⊙], **Kavitha Srinivasan**[2⊙], **Rengaraj Venkatesh**[2], **Ryan W. Buell**[3], **Kamalini Ramdas**[4]*

**1** ESMT Berlin, Berlin, Germany, **2** Aravind Eye Hospital, Thavalakuppam, Pondicherry, India, **3** Harvard Business School, Harvard University, Boston, Massachusetts, United States of America, **4** London Business School, London, United Kingdom

⊙ These authors contributed equally to this work.

* kramdas@london.edu

**Data Availability Statement:** As reported in our study protocol in the S2 Appendix, our study participants consented to have their potentially

## Abstract

In Shared Medical Appointments (SMAs), patients with similar conditions meet the physician together and each receives one-on-one attention. SMAs can improve outcomes and physician productivity. Yet privacy concerns have stymied adoption. In physician-deprived nations, patients' utility from improved access may outweigh their disutility from loss of privacy. Ours is to our knowledge the first SMA trial for *any* disease, in India, where doctors are scarce. In a 1,000-patient, single-site, randomized controlled trial at Aravind Eye Hospital, Pondicherry, we compared SMAs and one-on-one appointments, over four successive visits, for patients with glaucoma. We examined patients' satisfaction, knowledge, intention-to-follow-up, follow-up rates, and medication compliance rates (primary outcomes) using intention-to-treat analysis. Of 1,034 patients invited between July 12, 2016 –July 19, 2018, 1,000 (96.7%) consented to participate, and were randomly assigned to either SMAs ($N_{SMA}$ = 500) or one-on-one appointments ($N_{1-1}$ = 500). Patients who received SMAs showed higher satisfaction ($Mean_{SMA}$ = 4.955 (SD 0.241), $Mean_{1-1}$ = 4.920 (SD 0.326); difference in means 0.035; 95% CI, 0.017–0.054, p = 0.0002) and knowledge ($Mean_{SMA}$ = 3.416 (SD 1.340), $Mean_{1-1}$ = 3.267 (SD 1.492); difference in means 0.149; 95% CI, 0.057–0.241, p = 0.002) than patients who received one-on-one appointments. Across conditions, there was no difference in patients' intention-to-follow-up ($Mean_{SMA}$ = 4.989 (SD 0.118), $Mean_{1-1}$ = 4.986 (SD 0.149); difference in means 0.003; 95% CI, -0.006–0.012, p = 0.481) and actual follow-up rates ($Mean_{SMA}$ = 87.5% (SD 0.372), $Mean_{1-1}$ = 88.7% (SD 0.338); difference in means -0.012; 95% CI, -0.039–0.015, p = 0.377). Patients who received SMAs exhibited higher medication compliance rates ($Mean_{SMA}$ = 97.0% (SD 0.180), $Mean_{1-1}$ = 94.9% (SD 0.238); difference in means 0.020; 95% CI, 0.004–0.036, p = 0.013). SMAs improved satisfaction, learning, and medication compliance, without compromising follow-up rates or measured clinical outcomes. Peer interruptions were negatively correlated with patient

identifying data shared only with the researchers from London Business School and Harvard Business School. Any deviations from the current protocol requires approval of the Institutional Ethics Committee, Aravind Eye Hospital, Thavalakuppam, Pondicherry. The de-identified dataset supporting this research can be requested from the Institutional Ethics Committee, Aravind Eye Hospital, Thavalakuppam, Pondicherry (pdy. irb-office@pondy.aravind.org).

**Funding:** This work was supported by the Wheeler Institute at London Business School (WIBAD Ramdas_Sonmez CFP19 to NS, KR), the Institute of Entrepreneurship and Private Capital at London Business School (IIE_3432_2019 to NS, KR), Aravind Eye Hospital (KS, RV), and Harvard Business School (RB). The funders did not play any role in the study design, data collection and analysis, decision to publish, or preparation of the manuscript.

**Competing interests:** The authors have declared that no competing interests exist.

satisfaction in early-trial SMAs and positively correlated with patient satisfaction in later-trial SMAs.

**Trial registration**: The trial was registered with Clinical Trials Registry of India (https://ctri.nic.in/) with reference no. REF/2016/11/012659 and registration no. CTRI/2018/02/011998.

## Introduction

The demand for healthcare worldwide is soaring and exceeds supply. Public healthcare provision in low and low-middle income countries (LLMICs) is especially scarce [1–3]. Yet in most LLMICs, access to private care offered by qualified providers is the privilege of the relatively wealthy. For example, India's poorest citizens resort to either informal healthcare providers [4, 5] or immensely-crowded government care facilities with long waits–often resulting in delayed care or no care at all [6].

A highly-cited review of care quality in LLMICs published over a decade ago called for ways to improve the quality of both publicly and privately-provided care, by improving care delivery and outcomes for the poor, and better managing the burden of chronic diseases [7]. Accordingly, physicians, healthcare administrators, and policy makers worldwide continue to seek and implement new service delivery models to improve the efficiency and efficacy of healthcare delivery. One such model, introduced over two decades ago in the United States, is the shared medical appointment (SMA) [8]. In an SMA, multiple patients with the same condition meet with the physician in a group, and each receives one-on-one attention in turn. In a one-on-one appointment, the physician shares both customized information, tailored to a patient's specific needs, and standardized information that is also relevant to other patients with the same condition. SMAs reduce the repetition of standardized information while enabling the entire group to hear the physician's answers to questions posed by individual patients. Thus, SMAs have the potential to facilitate learning, while cutting costs by reducing physician time per patient.

The productivity gains from SMAs have been shown to be considerable [8–10]. In "drop-in" shared medical appointments, a patient's peers may vary from one SMA to the next, and productivity increases are typically 300% or higher [8]. As an example, in the US, a routine one-on-one appointment for cardiac preventive care can take 30 minutes, whereas a cardiologist may see 8–10 patients for this purpose in a 90-minute SMA, a 300% productivity improvement [11]. The productivity gains from SMAs could fuel expanded access to public healthcare and might also lower the price-point for accessing care privately. Also, because physicians divide their time across SMAs and one-on-one appointments, the improved productivity in SMAs reduces waiting times even for patients who choose not to attend SMAs, resulting in waiting time reductions of as much as 70% [10]. Moreover, a growing body of evidence from high-income countries (HICs) suggests that SMAs can yield superior medical outcomes for a wide variety of conditions, including neonatal care, asthma, rheumatoid arthritis, cardiovascular disease, and most notably type-2 diabetes, in both primary and secondary care settings [12, 13]. Conducive to their implementation, NHS England & Improvement in the UK offers training in the use of SMAs in primary care [14]. In the US, payors offer highly favourable reimbursement rates for SMAs [15]. SMAs have been implemented successfully for chronic disease follow up and for other non-urgent care [13].

Nevertheless, despite evidence of their efficiency and efficacy, availability of training and favourable reimbursement rates, SMAs adoption in HICs–except by a few leading providers,

e.g., Kaiser Permanente and the Cleveland Clinic in the United States–has been extremely limited [13]. Doctors and patients alike fear that the loss of privacy that is inherent to shared appointments may hinder the discussion of sensitive medical issues [16], and dampen patients' learning, satisfaction, and engagement [17]. More time with the doctor during one-on-one appointments is known to improve patients' knowledge and satisfaction [18], but little is known about how patients' experience and behaviour is impacted when a physician's time is shared among a group of patients [12].

In HICs, when SMAs are offered, patients are typically choosing between an SMA available at short notice vs. a one-on-one appointment on a later date. Since SMAs are for non-urgent care, patients may prefer to wait–to avail of the privacy of a one-on-one visit. In stark contrast, if SMAs were offered in poorer subpopulations in LLMICs, patients would likely face a very different choice: attending an SMA vs. receiving *no qualified care at all*. Faced with this trade-off, disutility from the loss of privacy may be outweighed by the utility gains from access to care. However, there is virtually no adoption of, or research on, SMAs in LLMICs. We searched MEDLINE and the Cochrane Library for articles on shared medical appointments (SMAs) published as of Nov. 26, 2022, using combinations of search terms for systematic reviews, shared medical appointments, group appointments or group visits. We also searched for articles on SMAs from Asia, Africa and Latin America, and for articles on SMAs for eye disease. (See S1 Appendix page 2 for search details.) To our knowledge, only 7 randomized controlled trials have evaluated SMAs across Asia, Africa, and Latin America–where healthcare demand far exceeds its supply–with none in India, a country with almost a fifth of the world's population, that spends only 1.1% of its gross domestic product (GDP) on health and faces a dire shortage of healthcare capacity [2]. In 2021, the Chief Scientist of the World Health Organization (WHO) called for research on SMAs in the developing world [19].

Although SMAs have received no attention in India, recent community-based research in India suggests that shared care delivery is both feasible and promising. Community health worker (CHW)–led group-based education and monitoring has been shown to reduce blood pressure [20]; facilitator-supported women's groups for perinatal care in Mumbai slums found evidence of behaviour change, despite no population-level effects on health care or mortality [21]; and a community-based peer-support lifestyle intervention led to type-2 diabetes prevention by lay peers while significantly improving some cardiovascular risk factors and physical functioning [22].

To study the effect of shared care delivery in the context of medical appointments, we conducted, to our knowledge, the first randomized controlled trial of SMAs for any disease in India, which is also only the 7th trial of SMAs across Asia, Africa and Latin America. Ours is also, to our knowledge, the first SMAs trial for glaucoma–which is a chronic condition that requires regular follow-up appointments and the second biggest cause of blindness worldwide. It is also distinguished among prior pragmatic randomized controlled trials of SMAs by its scale (most SMA trials have investigated fewer patients), its single-site nature (most large SMA trials have been conducted across multiple sites), and the extent to which, by design, aspects of the patient experience beyond the composition of the appointment itself have been carefully controlled. Our goal was to assess how SMAs affect two important aspects of patients' experience (knowledge gained and satisfaction) and two key behavioural outcomes (follow-up rates and medication compliance rates), while also examining two clinical outcomes (change in intraocular pressure ($\Delta$IOP) and change in optic nerve head cup-to-disc ratio ($\Delta$ONH)) for patients with primary glaucoma as secondary outcomes. Additionally, we examined physician time per patient and aimed to explore how patients' perceptions of shared appointments changed as they gained more experience with this format, by analysing verbal and nonverbal engagement measures obtained via video recording of every appointment.

## Methods

### Ethics statement

This study was approved by the ethics review boards at the Aravind Eye Hospital, Pondicherry, London Business School and Harvard Business School. All study participants provided written informed consent prior to participating in the study. The trial was registered with Clinical Trials Registry of India with reference no. REF/2016/11/012659 and registration no. CTRI/2018/02/011998. The trial protocol and amendments can be found in the supporting information (S2 Appendix).

### Study design

We conducted a large-scale, single-site, randomized controlled trial at the Aravind Eye Hospital in Pondicherry, India, which is one of 6 tertiary eye hospitals in the Aravind Eye Care System (AECS) in Southern India. AECS has served as a WHO Collaborating Centre, serves 4.5 million patients a year and trains over 300 hospitals across Africa and Asia in methods to deliver high-quality, low-cost care [23–25]. The Aravind Eye Hospital, Pondicherry's glaucoma clinic, where we conducted our trial, has five glaucoma consultants and four glaucoma fellows who collectively serve approximately 180 patients per day. In March 2023, the Aravind Eye Hospital, Pondicherry glaucoma clinic's median time from registration to exit was 237 minutes, median waiting time was 154 minutes, and median consultation time was 6 minutes. In our trial, 1,000 patients with primary glaucoma were randomly assigned to either attend SMAs with four other patients ($N^{SMA}$ = 500), or one-on-one appointments ($N^{1-1}$ = 500), during four successive routine follow-up visits scheduled four months apart. Enrolment occurred over two years.

### Participants

A glaucoma specialist on our research team reviewed the medical records of incoming patients to identify those who were eligible for our study. Eligible patients had primary glaucoma with no other vision-threatening conditions and had undergone no more than one surgery in each eye. Patients who had undergone a tube/shunt surgery in the past, were monocular, were likely to require surgical intervention soon, or were part of another trial, were excluded.

Eligible patients were invited to enrol in the trial. S1 Table provides a comparison of the demographics of the 34 patients who declined to participate with those of the 1,000 patients who enrolled.

### Randomisation and masking

Enrolled patients were randomly assigned in groups of five to either the treatment arm (SMAs) or the control arm (one-on-one appointments) by means of a random number generator, which ensured allocation concealment [26]. Throughout the trial, medical appointments for patients in the treatment and control arms were scheduled to occur at different times, so as to avoid bias due to patients in one condition observing the experiences of patients in the other condition. As ours was a service delivery intervention, it was not possible to mask patients and study doctors on our research team to group assignment.

### Procedures

Outside of our intervention, there were no systematic differences in the number or nature of physician interactions, nor in any other aspect of patient care [12]. Upon arriving in the Glaucoma Clinic, all patients met with a mid-level ophthalmic professional and a junior glaucoma

consultant, and underwent all or some of the following steps, depending on their disease stage and care requirements: patient medical history review, vision check, eye pressure check, field analysis and preliminary testing, dilation, and optical coherence tomography. After these initial steps, patients were escorted to a waiting room to await their medical appointments with one of the investigators (SK, RV). Both of the investigators were fellowship-trained glaucoma consultants with more than 10 years of experience who are well versed with the local language. The selection of these two investigators balanced care experience across providers and enabled us to avoid language barriers.

Our intervention determined whether groups of patients whose appointments were scheduled proximally were seen separately (in consecutive one-on-one appointments) or together (in an SMA). During their appointments, patients in the control arm (who received one-on-one appointments) experienced an eye examination, before receiving individualized recommendations from the doctor. Patients in the treatment arm (who received shared medical appointments) entered the appointment room together and waited their turn to experience an eye examination, before receiving individualized recommendations from the doctor. Patients experienced SMAs in groups of five during their first appointments and concluded their visit by scheduling their next appointment with a study coordinator. If the need arose due to changes in a patient's or a doctor's availabilities, the coordinator would reschedule the patient, as well as other enrolled patients, to fill in gaps in groups. Managing patients in both arms in groups enabled us to use identical scheduling and reminder procedures to ensure consistent experiences beyond the experimental manipulation. Nevertheless, due to such rescheduling, group size in follow-up SMAs varied from 2–6 patients (Mean = 4.262, SD = 0.991). Patients in both arms were welcome to ask questions at any time during their appointments. In one-on-one appointments, questions arose as a natural extension of the conversation between the doctor and the patient. In SMAs, after administering recommendations to the last patient, the doctor asked whether anyone in the group had questions. In both arms, an appointment ended naturally when there were no more questions.

All data–including physician notes, demographic information on each patient: age, gender, urban or rural residence, and education level, as well as medical information: glaucoma type, and the existence of relevant comorbidities–were recorded in participating patients' case report forms as described in the protocol provided in the supporting information (S2 Appendix). After each appointment, patients in both arms responded to a survey with questions assessing their satisfaction with the appointment, their knowledge about glaucoma, and their intention to return for a follow-up appointment (see S2 Table).

At the conclusion of the survey, patients were scheduled for their next appointment. An identical procedure was used to schedule or reschedule follow-up appointments for patients in both arms. One week and again one day prior to each appointment, the study coordinators called patients in both arms to remind them of their upcoming appointment.

Every appointment was videotaped with consent using a wall-mounted camera. The total time spent by the doctor on each patient group (in an SMA or in consecutive one-on-one appointments), and interruption measures, were obtained from this recording.

## Outcomes

For each patient, we examined four primary outcomes: satisfaction, knowledge, patient-reported medication compliance rate, and follow-up rate. Our secondary outcomes were two key medical outcomes–change in intraocular pressure ($\Delta$IOP) and change in optic nerve head cup-to-disc ratio ($\Delta$ONH) over 4 visits–appointment durations, as well as the number of verbal interruptions and gestures made by peers during each patient's examination.

Patient satisfaction and knowledge levels were measured at the end of each appointment. We recorded patient satisfaction levels on multiple dimensions (satisfaction with the appointment, with the degree to which their doubts were addressed, with their levels of learning, and with the degree to which they understood instructions) using a 5-point scale. We assessed each patient's knowledge level about glaucoma by administering a 5-question multiple choice test.

To assess intended and actual follow-up rates, at the end of each appointment we asked patients to report their intention to return for a follow-up visit. We subsequently tracked their follow-up behaviour, in particular, whether they returned within 30 days of their next scheduled appointment.

During each appointment, medication compliance rates were reported by each patient to the doctor. Doctors subsequently assessed each patient's condition through an eye examination by measuring intraocular pressure (IOP) and optic nerve head cup-to-disc ratio (ONH) levels. Since these measures tend to evolve slowly over time, ΔIOP (in mm Hg) and ΔONH (in mm) were calculated as the change in each measure from the first to the last appointment in the trial, which was 12 months by design (Mean = 13.211, SD = 1.855). For each patient, demographics, glaucoma type and systemic comorbidities were recorded at the start of the trial. All survey and knowledge assessment questions are included in S2 Table.

Videos of the appointments in both arms were transcribed and decoded to obtain measures of verbal and non-verbal engagement dynamics among the patients and physicians, as well as to measure appointment durations. Appointment durations were used to estimate physician time per patient.

Patient health and safety were continually monitored by the physicians engaged in the trial. As our trial involved no direct medical interventions, no adverse events were identified.

## Statistical analysis

As we mention in our study protocol in the S2 Appendix, we determined the sample size for our trial by using effect sizes in line with prior research on shared medical appointments [27], and conducting power analyses using data gathered prior to the start of our trial, on knowledge levels and follow-up rates of patients who experienced pilot SMAs and regular one-on-one appointments.

Using a 10-point knowledge survey administered to 50 patients attending pilot SMAs and 50 patients attending regular one-on-one appointments prior to the start of our trial, the mean knowledge in the SMAs arm was 6.68 (SD = 2.00) and the mean knowledge in the one-on-one arm was 6.18 (SD = 2.07) using the power method [28]. The mean difference in knowledge levels was 0.5. Using $\alpha = 0.01$ and $\beta = 0.10$, we estimated the required sample size to be 493 patients in each arm. Using a similar method, we also examined follow up rate for patients in pilot SMAs and regular one-on-one appointments prior to our trial start and obtained a similar sample size. Based on these analyses we chose an initial target sample size of 1,000 (with 500 patients in each condition). As we had specified that we would do in our protocol, we used trial data collected in the first month of the trial to further calibrate the sample size and decided to continue with 1,000 as our target sample size (500 in each arm). Continuous outcomes were analysed using linear regression and binary outcomes were analysed using logistic regression. For both regression types, dependent variables were patient outcomes and the predictor was a dummy variable that denoted whether the patient was a part of the SMA treatment. We present our outcomes as predicted means (SD) for both trial arms. Standard errors were clustered at the patient level. Results are presented as mean differences. We note and show in S3 Table that all results are substantively similar when we control for the patient's biological sex, age, urbanity, the presence of comorbidities, and education level, as well as an indicator variable

denoting the identity of the doctor. For ordinal outcomes, while for ease of exposition we used linear regression, we acknowledge that such models have some limitations. Our results are robust to the use of generalized ordered logit regressions for the treatment effect on our ordinal outcomes (see S32–S37 Tables) [29].

Five different subgroup analyses were performed based on patients' sex, age, urban vs. rural residence, education, and comorbidities. The treatment effect within each subgroup was examined by including appropriate treatment-by-subgroup interaction variables in the regression models. Pairwise comparisons of the predicted means (SD) were reported after running these regressions to calculate the treatment effect within each subgroup with a 95% confidence interval. A chi-squared test was used to determine whether the effects of SMAs and one-on-one appointments differed across subgroups in each analysis. We note, and show in S4–S22 Tables that all results are substantively similar when we control for the patient's biological sex, age, urbanity, the presence of comorbidities, and education level, as well as an indicator variable denoting the identity of the doctor. All analyses were generated using STATA software version IC 14.2.

As patient safety was not assessed to be compromised by this intervention, a data monitoring committee was not formed to oversee this study. Our trial was registered through Clinical Trial Registry in India with reference number REF/2016/11/012659 and registration number CTRI/2018/02/011998.

### Inclusivity in global research

Additional information regarding the ethical, cultural, and scientific considerations specific to inclusivity in global research is included in the Supporting Information (S1 Text).

## Results

From July 12, 2016 through July 19, 2018, a total of 1,034 patients were identified as being eligible for the trial and were invited to participate. Of those invited, 1,000 (96.7%) consented to participate (Fig 1). We analyse the data from these patients, 500 (50%) of whom experienced SMAs and 500 (50%) of whom experienced one-on-one appointments. Details of participants' baseline characteristics in each group are provided in Table 1. Of our participants, 38% lived in rural areas, over half had at most secondary school education, 42% were female and 37% were diabetic.

The results for each primary outcome are provided in Table 2. Relative to patients who experienced one-on-one appointments, patients who received SMAs reported significantly higher levels of satisfaction with their appointments on three out of four measured dimensions, exhibited higher levels of knowledge about glaucoma, and reported higher medication compliance rates.

On a 5-point scale, patients' overall satisfaction with the appointment was 4.955 (SD 0.241) among patients who experienced SMAs and 4.920 (SD 0.326) among patients who experienced one-on-one appointments (difference in means 0.035; 95% CI, 0.017–0.054, p = 0.0002). Patients who experienced SMAs also reported higher satisfaction with the degree to which their doubts were addressed (difference in means 0.036; 95% CI, 0.019–0.054, p = 0.0001) and with their levels of learning (difference in means 0.090; 95% CI, 0.059–0.121, p<0.0001). Patients reported an insignificant difference in satisfaction with understanding instructions across SMAs and one-on-one appointments (difference in means 0.006; 95% CI, -0.007–0.018, p = 0.362).

Consistent with their higher reported satisfaction with the degree to which their doubts were addressed and their levels of learning, patients in SMAs exhibited higher knowledge

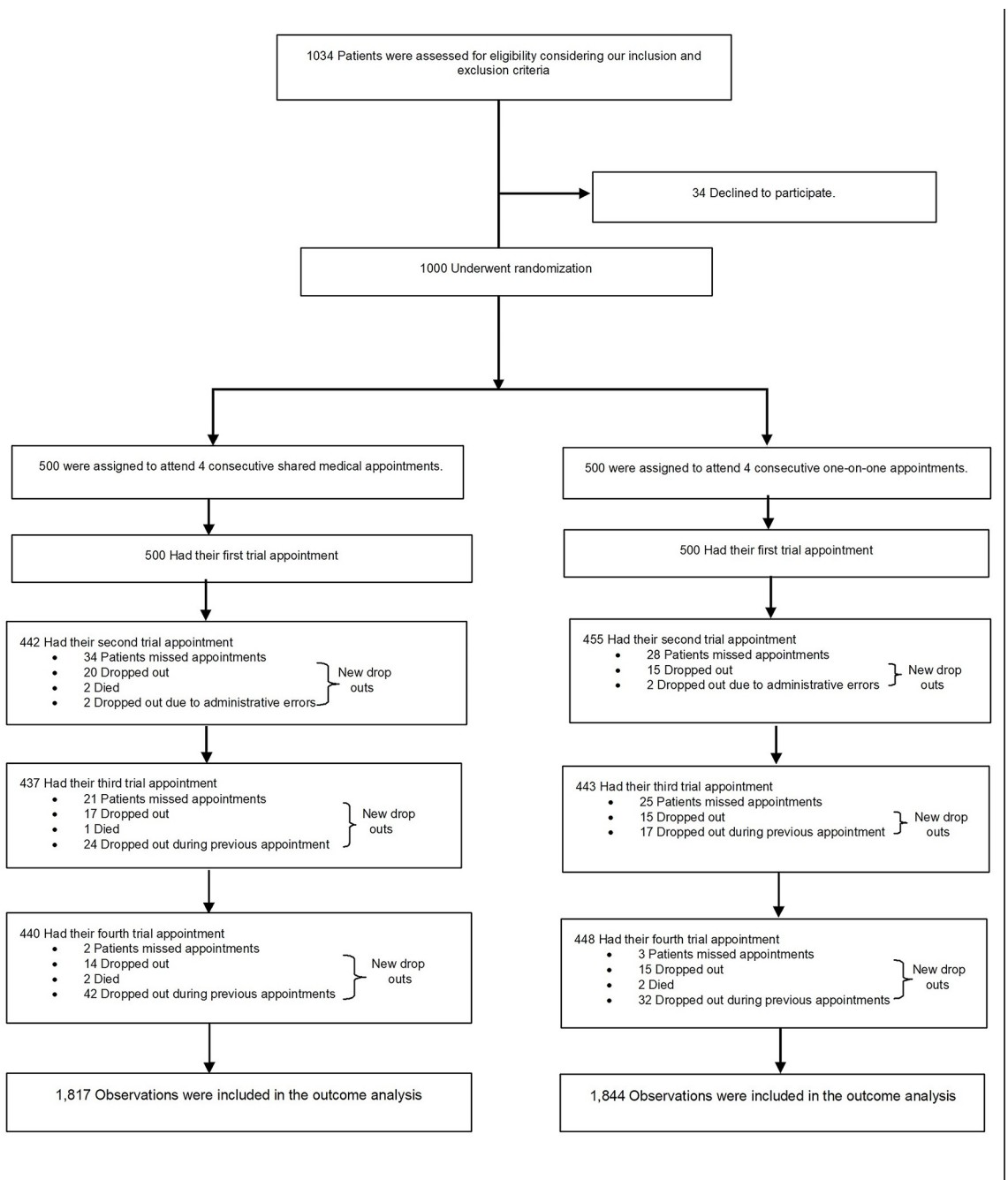

**Fig 1. Trial profile.**

levels, correctly answering an average of 3.416 (SD 1.340) questions out of five, relative to an average of 3.267 (SD 1.492) correct answers for patients in one-on-one appointments (difference in means 0.149; 95% CI, 0.057–0.241, p = 0.002). Moreover, the patient-reported medication compliance rate was significantly higher for patients who experienced SMAs: 0.970 (SD 0.180) in SMAs and 0.949 (SD 0.238) in one-on-one appointments (difference in means 0.020; 95% CI, 0.004–0.036, p = 0.013). In Fig 2 we report the average noncompliance behaviour of

**Table 1. Demographics and baseline characteristics.**

| | SMA (n = 500) | One-On-One (n = 500) |
|---|---|---|
| **Dependent Variables¶** | | |
| Medication Compliance Rate | 0.931 (0.252) | 0.929 (0.255) |
| Probability of Returning within 30 Days‡ | 0.696 (0.460) | 0.700 (0.458) |
| Intra Ocular Pressure (IOP) | 15.352 (3.400) | 15.198 (3.280) |
| Optic Nerve Head Cup-to-Disk Ratio (ONH) | 0.673 (0.142) | 0.672 (0.144) |
| **Demographic Variables:** | | |
| Age (SD)¶ | 61.928 (9.160) | 62.150 (9.514) |
| Proportions of Male Patients | 288 (57.6%) | 314 (62.8%) |
| Urban | 308 (48.7%) | 304 (60.8%) |
| Education Level† | 2.548 (1.167) | 2.588 (1.253) |
| **Medical Variables:** | | |
| **Proportion of Glaucoma Types** | | |
| Primary Open Angle Glaucoma (POAG) | 366 (73.2%) | 374 (74.8%) |
| Primary Angle Closure Disease (PACD) | 113 (22.6%) | 111 (22.2%) |
| Ocular Hypertension (OHT) | 5 (1%) | 5 (1%) |
| Pseudoexfoliation Glaucoma (PXF Glaucoma) | 16 (3.2%) | 10 (2%) |
| **Proportion of Comorbidities** | | |
| Diabetes | 184 (36.8%) | 189 (37.8%) |
| Hypertension | 176 (35.2%) | 189 (37.8%) |
| Cardiac Disease | 20 (4%) | 17 (3.4%) |
| Asthma / Chronic Obstructive Pulmonary Disease (COPD) | 11 (2.2%) | 8 (1.6%) |
| Other Chronic Diseases | 2 (0.4%) | 5 (1%) |

¶ Data are mean (SD).

‡ For Probability of Returning within 30 days, we used archival data from the appointment preceding the trial since by construction, every patient was on time for the first trial appointment.

† The Education variable is scaled as: Illiterate (1); Primary School Education (2); Secondary School Education (3); Undergraduate Education (4); Postgraduate Education (5).

patients in both arms, during the four appointments in our trial. There were no significant differences in intended or actual follow-up rates, nor were there significant differences in ΔIOP or ΔONH during the period of our study. Results for these secondary outcomes are in S5–S8 Tables. S23–S30 Tables demonstrate baseline medication compliance rates, probability of returning within 30 days of the scheduled appointment date, intraocular pressure, and optic nerve head cup-to-disc ratio levels in both arms when the trial started. Physician time per patient was 3.446 minutes (SD 0.803) in the treatment arm and 3.041 minutes (SD 0.690) in the control arm (difference in means 0.405; 95% CI, 0.356–0.454, p<0.0001).

Across five subgroup analyses for each of the four primary outcomes, SMAs were in no case detrimental to a subgroup. Analyses that showed statistically significant differences across subgroups are described below. We observed significant subgroup effects for the medication compliance rate in two of five subgroup analyses (see Table 3). For female patients, the medication compliance rate was 0.987 (SD 0.126) in SMAs and 0.943 (SD 0.260) in one-on-one appointments (difference in means 0.044; 95% CI, 0.019–0.069, p = 0.001), whereas for male patients, it was 0.957 (SD 0.209) in SMAs and 0.953 (SD 0.224) in one-on-one appointments (difference in means 0.004; 95% CI, -0.017–0.025, p = 0.007 for the interaction between SMAs and sex); for patients aged 65 or younger it was 0.975 (SD 0.161) in SMAs and 0.939 (SD 0.271) in one-on-one appointments (difference in means 0.036; 95% CI, 0.014–0.057, p = 0.001), whereas for

**Table 2. Primary outcomes.**

| Primary Outcomes | SMA | One-On-One | Difference in Means (95% CI) ⚲ | p value |
|---|---|---|---|---|
| Satisfaction with the Appointment | 4.955 (0.241) | 4.920 (0.326) | 0.035 (0.017–0.054) | 0.0002 |
| Satisfaction with Doubts Addressed | 4.975 (0.194) | 4.939 (0.330) | 0.036 (0.019–0.054) | 0.0001 |
| Satisfaction with Learning | 4.899 (0.357) | 4.810 (0.579) | 0.090 (0.059–0.121) | <0.0001 |
| Satisfaction with Understanding Instructions | 4.980 (0.175) | 4.974 (0.205) | 0.006 (-0.007–0.018) | 0.362 |
| Knowledge Level | 3.416 (1.340) | 3.267 (1.492) | 0.149 (0.057–0.241) | 0.002 |
| Medication Compliance Rate† | 0.970 (0.180) | 0.949 (0.238) | 0.020 (0.004–0.036) | 0.013 |
| Intention to Return | 4.989 (0.118) | 4.986 (0.149) | 0.003 (-0.006–0.012) | 0.481 |
| Probability of Returning within 30 Days† | 0.875 (0.372) | 0.887 (0.338) | -0.012 (-0.039–0.015) | 0.377 |

Data are mean (SD).

⚲ Continuous outcomes were analysed by means of linear regression. Binary outcomes were analysed by means of logistic regression. 95% confidence intervals were constructed with errors clustered at the patient level.

† We examined data from 2nd, 3th and 4th appointment of the trial for Medication Compliance Rate and Probability of Returning within 30 Days outcomes. For these outcomes, values in the 1st appointment are treatment independent. For other measures, we use data from all four trial appointments.

patients over 65 years old, it was 0.961 (SD 0.208) in SMAs and 0.965 (SD 0.177) in one-on-one appointments (difference in means -0.004; 95% CI, -0.028–0.020, p = 0.019 for the interaction between SMAs and age subgroup). Subgroup analyses can be found in S9–S22 Tables for all other primary outcomes and in S5–S8 Tables for all secondary outcomes. No adverse events were attributed to the trial.

## Discussion

In this randomized controlled trial, which is, to our knowledge, the first trial of SMAs in India, and first SMAs trial worldwide for glaucoma, we examined the performance of SMAs relative to one-on-one appointments for regular follow up of patients with glaucoma, at the Aravind Eye Hospital, Pondicherry. We find that relative to one-on-one appointments, SMAs

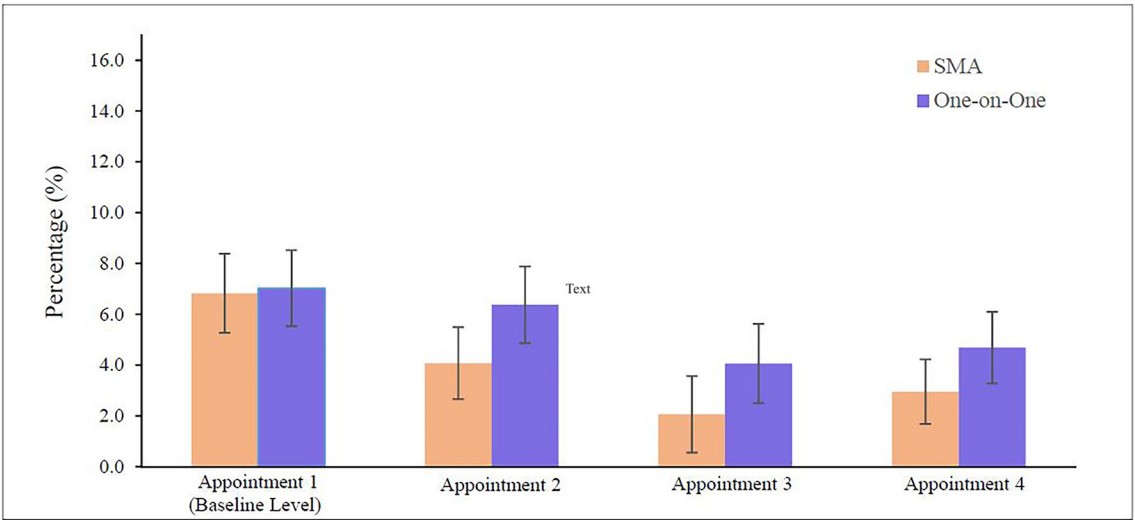

**Fig 2. Percentage of noncompliant patients by trial appointment.** For each trial appointment, bars represent the percentage (%) of noncompliant patients in SMA and one-on-one arm.

**Table 3. Medication compliance rate in prespecified subgroups.**

| | SMA | One-On-One | Difference (95% CI) ¶ | p value for Interaction |
|---|---|---|---|---|
| **Prespecified Subgroup‡** | | | | |
| **Sex** | | | | |
| Female | 0.987 (0.126) | 0.943 (0.260) | 0.044 (0.019–0.069)*** | 0.007 |
| (N^SMA = 555, N^1-1 = 494) | | | | |
| Male | 0.957 (0.209) | 0.953 (0.224) | 0.004 (-0.017–0.025) | |
| (N^SMA = 764, N^1-1 = 852) | | | | |
| **Location** | | | | |
| Rural | 0.965 (0.188) | 0.928 (0.289) | 0.038 (0.008–0.067)** | 0.252 |
| (N^SMA = 519, N^1-1 = 540) | | | | |
| Urban | 0.972 (0.175) | 0.964 (0.193) | 0.008 (-0.010–0.027) | |
| (N^SMA = 800, N^1-1 = 806) | | | | |
| **Education Level** | | | | |
| Illiterate | 0.942 (0.279) | 0.885 (0.354) | 0.058 (-0.014–0.129) | 0.271 |
| (N^SMA = 139, N^1-1 = 165) | | | | |
| Primary School | 0.967 (0.182) | 0.964 (0.188) | 0.003 (-0.015–0.022) | |
| (N^SMA = 785, N^1-1 = 746) | | | | |
| Secondary School | 0.963 (0.183) | 0.951 (0.207) | 0.012 (-0.054–0.079) | |
| (N^SMA = 54, N^1-1 = 81) | | | | |
| Undergraduate | 0.986 (0.116) | 0.958 (0.221) | 0.028 (-0.009–0.065) | |
| (N^SMA = 213, N^1-1 = 168) | | | | |
| Postgraduate | 0.992 (0.088) | 0.941 (0.288) | 0.051 (0.007–0.095)** | |
| (N^SMA = 128, N^1-1 = 186) | | | | |
| **Age** | | | | |
| ≤65 | 0.975 (0.161) | 0.939 (0.271) | 0.036 (0.014–0.057)*** | 0.019 |
| (N^SMA = 830, N^1-1 = 802) | | | | |
| >65 | 0.961 (0.208) | 0.965 (0.177) | -0.004 (-0.028–0.020) | |
| (N^SMA = 489, N^1-1 = 544) | | | | |
| **Comorbidities** | | | | |
| Diabetes | 0.974 (0.156) | 0.949 (0.208) | 0.024 (0.002–0.047)** | 0.307† |
| (N^SMA = 496, N^1-1 = 513) | | | | |
| Hypertension | 0.969 (0.192) | 0.963 (0.182) | 0.006 (-0.018–0.030) | |
| (N^SMA = 456, N^1-1 = 516) | | | | |
| Cardiac Disease | 0.941 (0.233) | 0.959 (0.192) | -0.018 (-0.102–0.065) | |
| (N^SMA = 51, N^1-1 = 49) | | | | |
| Asthma / Chronic Obstructive Pulmonary Disease (COPD) | 0.923 (0.247) | 0.952 (0.208) | -0.029 (-0.159–0.101) | |
| (N^SMA = 26, N^1-1 = 21) | | | | |
| Other Chronic Diseases† | 1.000 (0.000) | 0.857 (0.363) | n/a | |
| (N^SMA = 6, N^1-1 = 14) | | | | |
| **Overall** | 0.970 (0.180) | 0.949 (0.238) | 0.020 (0.004–0.036)** | |
| (N^SMA = 1319, N^1-1 = 1346) | | | | |

Data are mean (SD).

‡ In each row, the sample sizes $N^{SMA}$ and $N^{1-1}$ denote the number of observations–across all relevant appointments–at the subgroup level in question (e.g., Female or Male), in SMAs and 1-1s respectively.

¶ Medication Compliance Rate was analysed by means of logistic regression. 95% confidence intervals were constructed using the errors clustered at patient level.

*** $p<0.01$, ** $p<0.05$, * $p<0.1$ –these p values are associated with the treatment effect within each subgroup.

† Due to lack of outcome variation in some of the subgroups, it was only possible to calculate the chi-square p value for the interaction using the subgroups for which we could derive difference and confidence intervals from regression models. Mean (SD) derived from summary statistics when the model could not have been estimated due to lack of variation in one or two arms of one subgroup and resulted in n/a as the difference in means.

significantly improved two important patient-experience-related outcomes–patient satisfaction and knowledge, and one key behavioural outcome–medication compliance rate, with no significant difference in a second key behavioural outcome–follow-up rate. Patients in both arms were equally stable during the trial, based on two medical outcomes–ΔIOP and ΔONH. Physician time per patient was higher in the treatment arm.

Patients who were randomly assigned to receive SMAs reported higher levels of satisfaction with their appointments, with their doubts being addressed and with their levels of learning. It is important to note that these significant improvements in satisfaction were detected despite high baseline satisfaction levels in the control group across all measured dimensions (*Mean*>4.807/5.000 in all cases). The significant improvement in medication compliance rates arising from SMAs was similarly striking. Despite a high baseline medication compliance rate of 94.9% (SD 0.238) for patients who received one-on-one appointments, a 39% reduction in non-compliance was observed among patients who were randomly assigned to receive SMAs. Fig 2 shows that after the first trial appointment, the percentage of non-compliant patients fell significantly in both arms. We found that patients who were assigned to receive SMAs exhibited significantly lower non-compliance rates in trial appointments 2–4 (because non-compliance is measured at the start of each appointment, non-compliance in trial appointment 1 was not impacted by our intervention). The improvement in non-compliance in the treatment arm suggests that SMAs may be well-suited to increase medication compliance in developing countries, where compliance with medication is known to be lower than in developed countries [30]. Future work can also delve into the effect sizes of SMAs on other patient outcomes.

The large sample size in this trial enabled additional investigations of the primary outcomes through five subgroup analyses, based on sex, urban vs. rural residence, age, education level, and comorbidities. We find that SMAs differentially impact medication compliance rates for subgroups based on sex and age. Specifically, our subgroup analyses reveal that the beneficial impact of SMAs on medication compliance rates is significantly greater for female patients. That women benefited from SMAs is heartening considering that India ranks 132nd out of 146 countries on the World Economic Forum's global gender gap index [31], mortality and care access is lower for girls and women globally [32], and women are more likely than men to be sight impaired [33]. The beneficial impact of SMAs on medication compliance rates is also significantly greater for patients aged 65 or younger.

Although SMAs have been shown to improve efficiency and outcomes in HICs [12], loss of privacy has severely hampered their adoption [34, 35]. Our results indicate that in our Indian sample, patients' satisfaction is higher in shared appointments, and that through SMAs, patients' knowledge levels and medication compliance rates are improved. Our exit survey indicate that patients viewed SMAs as more fair than one-on-one appointments, felt more at ease in SMAs, and felt that the physician was more caring towards both them and their fellow patients, in SMAs (see S31 Table). Patients also perceived that the physician spent more time with them individually in an SMA (see S31 Table).

To further explore privacy concerns, we used video recordings (obtained with consent) of every trial SMA, to identify instances when a patient's one-on-one time with the doctor was interrupted by a peer's comment or gesture. S1 and S2 Figs, which plot patient satisfaction against interruptions, provide descriptive evidence that interruptions by their peers at first hurt patient satisfaction, but with experience of SMAs, patients grew to value such interruptive interactions.

In our trial, physician time per patient in the control arm was 3.04 minutes (vs. about 8–10 minutes in US-based glaucoma studies [36, 37]). The dramatically lower time per patient at our trial site reflects AECS's design philosophy: senior doctors perform only those tasks that cannot be performed by junior doctors or other staff [38]. Also, at the Aravind Eye Hospital,

Pondicherry, patients are given a follow-up appointment date, but no scheduled time slot. An appointment finishes when there are no more questions. Because patients in SMAs asked more questions, doctor time per patient was 13.32% more in SMAs than in one-on-one appointments. Importantly, however, our related research from the same trial indicates that at a duration of 80% of the average duration of five consecutive one-on-one appointments, SMAs resulted in the same number of questions asked [39]. This indicates that the Aravind Eye Hospital, Pondicherry glaucoma clinic could have improved information exchange while reducing physician time per patient, by shortening the duration of the SMAs. Also, in our trial we kept peers together over the trial appointments, to enable easier implementation of the experiment. In practice, "drop in" SMAs, in which the peers may vary from one appointment to the next, result in greater productivity gains by design [9].

We studied patients with glaucoma, a disease for which patients may be less concerned about privacy. Moreover, glaucoma appointments are quite protocol-based and therefore may be more amenable to a group format than appointments for other conditions. In HICs, SMAs have been successfully implemented for a wide range of conditions. Future research in LLMICs should investigate the effects of SMAs on the experiences and behaviours of patients with other diseases.

Although our results are robust and consequential, we cannot pinpoint with this randomised controlled trial the exact underlying mechanisms that drive observed differences in outcomes between SMAs and one-on-one appointments. Nevertheless, our findings that more doubts are addressed, and that greater learning occurs in SMAs, may contribute to the greater satisfaction and knowledge gained in SMAs, and to patients' increased willingness to adhere to prescribed medications. Future work could delve more deeply into the mechanisms underlying these relationships and explore the impact of SMAs in a broader array of conditions and contexts. In HICs SMAs have been studied most widely for type-2 diabetes, a disease with high prevalence with LLMICs, and for many other diseases pertinent to the world's poorest populations.

As many of the world's poorest billion reside in rural areas in LLMICs [2], one potentially promising application of SMAs is in telehealth. By enabling a lower yet profitable price point–as with in person SMAs–virtual SMAs could spur private provision of telehealth in rural areas [19]. Also, as online care delivery may feel isolating, bringing together groups of patients online may foster the satisfaction, knowledge, and compliance benefits observed in this study, while also providing social connection. Indeed, just as shared learning environments in classrooms around the world gave way to shared online learning in the shadow of the global COVID-19 pandemic, virtual SMAs might enhance the efficacy and efficiency of online patient care [14, 40], and we leave this promising area of inquiry to future scholars.

Taken together, this study suggests that SMAs can improve patient satisfaction, knowledge, and medication compliance, without compromising follow-up rates, change in intraocular pressure (ΔIOP) or change in optic nerve head cup-to-disc ratio (ΔONH) for patients with glaucoma–results which may help allay concerns about the viability of SMAs, spurring further consideration of this care delivery model worldwide, and in LLMICs in particular [15]. The high representation of less-educated, rural, diabetic, and female patients in our sample suggests that SMAs may be valued in underprivileged LLMIC populations.

More satisfied and knowledgeable patients are likely to engage more in their own health care, which is consistent with the higher medication compliance rate observed in our trial [41]. For chronic conditions, low medication compliance rates and low follow-up rates for regular appointments are known to catalyse adverse events that result in expensive hospitalizations and degraded patient quality of life. Because the documented improvements in health outcomes from SMAs in HICs are likely a result of patients' experience and behaviours, and

acceptance of SMAs may be culture-specific, further careful scientific evidence on how SMAs affect patient-experience-related outcomes and patient behaviours in other chronic disease contexts is needed for hospital systems and policy makers to encourage wider implementation of SMAs in primary and secondary care settings, in both HICs and LLMICs.

## Supporting information

**S1 Checklist. CONSORT 2010 checklist of information to include when reporting a randomised trial\*.**
(DOC)

**S1 Appendix. Supplementary appendix.**
(DOCX)

**S2 Appendix. Trial protocol and amendments.**
(PDF)

**S1 Text. Inclusivity in global research checklist.**
(DOCX)

**S1 Fig. Coefficient size of interruptions per minute regressed on patient satisfaction.**
(PDF)

**S2 Fig. Coefficient size of gestures per minute regressed on patient satisfaction.**
(PDF)

**S1 Table. Characteristics of patients who attended versus declined to participate.**
(PDF)

**S2 Table. Satisfaction and knowledge survey.**
(PDF)

**S3 Table. Primary outcomes with controls.**
(PDF)

**S4 Table. Medication compliance rate in prespecified subgroups with controls.**
(PDF)

**S5 Table. Change in intraocular pressure (ΔIOP), in prespecified subgroups.**
(PDF)

**S6 Table. Change in intraocular pressure (ΔIOP), in prespecified subgroups with controls.**
(PDF)

**S7 Table. Change in optic nerve head cup-to-disk ratio (ΔONH), in prespecified subgroups.**
(PDF)

**S8 Table. Change in optic nerve head cup-to-disk ratio (ΔONH), in prespecified subgroups with controls.**
(PDF)

**S9 Table. Satisfaction with the appointment, in prespecified subgroups.**
(PDF)

**S10 Table. Satisfaction with the appointment, in prespecified subgroups with controls.**
(PDF)

**S11 Table. Satisfaction with doubts addressed, in prespecified subgroups.**
(PDF)

**S12 Table. Satisfaction with doubts addressed, in prespecified subgroups with controls.**
(PDF)

**S13 Table. Satisfaction with learning, in prespecified subgroups.**
(PDF)

**S14 Table. Satisfaction with learning, in prespecified subgroups with controls.**
(PDF)

**S15 Table. Satisfaction with understanding instructions, in prespecified subgroups.**
(PDF)

**S16 Table. Satisfaction with understanding instructions, in prespecified subgroups with controls.**
(PDF)

**S17 Table. Patient knowledge, in prespecified subgroups.**
(PDF)

**S18 Table. Patient knowledge, in prespecified subgroups with controls.**
(PDF)

**S19 Table. Intention to return, in prespecified subgroups.**
(PDF)

**S20 Table. Intention to return, in prespecified subgroups with controls.**
(PDF)

**S21 Table. Probability of returning within 30 days of the scheduled appointment date, in prespecified subgroups.**
(PDF)

**S22 Table. Probability of returning within 30 days of the scheduled appointment date, in prespecified subgroups with controls.**
(PDF)

**S23 Table. Baseline medication compliance rate, in prespecified subgroups.**
(PDF)

**S24 Table. Baseline medication compliance rate, in prespecified subgroups with controls.**
(PDF)

**S25 Table. Baseline probability of returning within 30 days of the scheduled appointment date level, in prespecified subgroups.**
(PDF)

**S26 Table. Baseline probability of returning within 30 days of the scheduled appointment date level, in prespecified subgroups with controls.**
(PDF)

**S27 Table. Baseline intraocular pressure (IOP) level, in prespecified subgroups.**
(PDF)

## Acknowledgments

The authors thank Saraswathi Udhayakumar, Tamilarasi Murugan, Kalaiselvi Kaliyamurthy and Sindhu Natarajan for their care and dedication in administering the trial, and Melissa Ouellet and Ibrahim Ata for their invaluable research assistance. We also thank the Wheeler Institute for Business and Development at London Business School, and the Institute of Innovation, and Entrepreneurship at London Business School for their generous financial support. All errors remain our own.

## Author Contributions

**Conceptualization:** Nazlı Sönmez, Kavitha Srinivasan, Rengaraj Venkatesh, Ryan W. Buell, Kamalini Ramdas.

**Data curation:** Nazlı Sönmez, Kavitha Srinivasan, Ryan W. Buell, Kamalini Ramdas.

**Formal analysis:** Nazlı Sönmez, Ryan W. Buell, Kamalini Ramdas.

**Funding acquisition:** Nazlı Sönmez, Rengaraj Venkatesh, Ryan W. Buell, Kamalini Ramdas.

**Investigation:** Nazlı Sönmez, Kavitha Srinivasan, Rengaraj Venkatesh, Ryan W. Buell, Kamalini Ramdas.

**Methodology:** Nazlı Sönmez, Ryan W. Buell, Kamalini Ramdas.

 

**Project administration:** Kavitha Srinivasan, Rengaraj Venkatesh, Ryan W. Buell, Kamalini Ramdas.

**Resources:** Rengaraj Venkatesh, Ryan W. Buell, Kamalini Ramdas.

**Software:** Nazlı Sönmez, Ryan W. Buell, Kamalini Ramdas.

**Supervision:** Rengaraj Venkatesh, Ryan W. Buell, Kamalini Ramdas.

**Validation:** Nazlı Sönmez, Kavitha Srinivasan, Rengaraj Venkatesh, Ryan W. Buell, Kamalini Ramdas.

**Visualization:** Nazlı Sönmez, Ryan W. Buell, Kamalini Ramdas.

**Writing – original draft:** Nazlı Sönmez, Ryan W. Buell, Kamalini Ramdas.

**Writing – review & editing:** Nazlı Sönmez, Kavitha Srinivasan, Rengaraj Venkatesh, Ryan W. Buell, Kamalini Ramdas.

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
