## [Decision Letter · Decision Letter 0]

16 Mar 2023

PGPH-D-23-00116

Evidence from the First Shared Medical Appointments (SMAs) Trial in India:  SMAs Increase the Satisfaction, Knowledge, and Medication Compliance of Patients with Glaucoma

Dear Dr. Ramdas,

Thank you for submitting your manuscript to PLOS Global Public Health. After careful consideration, we feel that it has merit but does not fully meet PLOS Global Public Health’s publication criteria as it currently stands. Therefore, we invite you to submit a revised version of the manuscript that addresses the points raised during the review process.

As you can see below, I received two divergent reviews on your manuscript, with Reviewer #2 raising several concerns about the trial methodology and reporting. So, please ensure your revision adequately addresses their concerns.

We look forward to receiving your revised manuscript.

Kind regards,

Madhukar Pai, MD, PhD

Editor-In-Chief

Journal Requirements:

1. Please include a complete copy of PLOS’ questionnaire on inclusivity in global research in your revised manuscript. Our policy for research in this area aims to improve transparency in the reporting of research performed outside of researchers’ own country or community. The policy applies to researchers who have travelled to a different country to conduct research, research with Indigenous populations or their lands, and research on cultural artefacts. The questionnaire can also be requested at the journal’s discretion for any other submissions, even if these conditions are not met.  Please find more information on the policy and a link to download a blank copy of the questionnaire here: https://journals.plos.org/plosone/s/best-practices-in-research-reporting. Please upload a completed version of your questionnaire as Supporting Information when you resubmit your manuscript.

2. Please include an explanation for the retrospective CT registration and confirmation that all related CTs are registered.

3. During your revisions, please confirm whether the wording in the title is correct and update it in the manuscript file and online submission information if needed. Specifically, the article title should specify that the study was a randomised trial, in accordance with the CONSORT checklist. Please also change your manuscript article type to 'Clinical Trial' when you resubmit your manuscript.

4. Since your data is not available for proprietary reasons, please explain via email why the data is not available. Please also include the contact information for the third party organization that should be contacted should other researchers want to request access to this data and please include the full citation of where the data can be found. We also request that you verify with us via email that any researcher will be able to obtain the data set in the same manner that the you have obtained it. If you feel you are unwilling or unable to adhere to this policy, please explain your reasons by return email and your exemption request will be escalated to the editor for approval. Your exemption request will be handled independently and will not hold up the peer review process, but will need to be resolved should your manuscript be accepted for publication. One of the Editorial team will be in touch if they require more information.

5. Please provide separate figure files in .tif or .eps format and remove the embedded figures from the manuscript file.

Reviewers' comments:

Reviewer's Responses to Questions

**Comments to the Author**

1. Does this manuscript meet PLOS Global Public Health’s publication criteria? Is the manuscript technically sound, and do the data support the conclusions? The manuscript must describe methodologically and ethically rigorous research with conclusions that are appropriately drawn based on the data presented.

Reviewer #1: Yes

Reviewer #2: No

2. Has the statistical analysis been performed appropriately and rigorously?

Reviewer #1: Yes

Reviewer #2: No

3. Have the authors made all data underlying the findings in their manuscript fully available (please refer to the Data Availability Statement at the start of the manuscript PDF file)?

Reviewer #1: Yes

Reviewer #2: No

4. Is the manuscript presented in an intelligible fashion and written in standard English?

Reviewer #1: Yes

Reviewer #2: Yes

5. Review Comments to the Author

Reviewer #1: This is a very innovative and interesting study. Overall the paper is well written and well structured. There is adequate context to introduce the topic of SMA and a good description of methodology.

Comments on Number of patients in a SMA group

• Line 162 mentioned that participants will be attending with 4 other participants (the mean value), whereas Line 203 mentions 5 and then 2 to 6. It maybe better to clarify in Line 162 that this is the mean value.

• Can you provide justification for this grouping? Why were they grouped only with 2 to 6 patients? Why not more?

• Was there a difference in the experience and knowledge between groups that had 2 members vs groups that had 6 members?

Comments on Number of investigators

• Can you provide justification for choosing only 2 investigators in Line 193?

• Was there a difference in the knowledge/satisfaction among patients seen by SK vs those seen by RV?

Others:

The use of ‘Aravind’ in paragraphs from Line 436 to 448 gives the impression that you are talking about the entire Hospital (Aravind Eye Hospital) rather than ‘the trial’/’our study’

Reviewer #2: In the Introduction section, the authors suggest that in sub-Saharan Africa, half of the healthcare provision is private, and the poorest billion people incur catastrophic out-of-pocket healthcare expenses. The study site, Puducherry, is a relatively affluent site, and the results obtained from very poor sub-Saharan settings cannot be extrapolated to Puducherry. Furthermore, the authors investigated glaucoma, a disease that does not typically result in catastrophic healthcare expenses. Additionally, the authors suggest that even in Sub-Saharan Africa, half of the healthcare is provided by the private sector, and they also mention poorly staffed and crowded government outpatient departments where patients are often turned away without receiving treatment. However, this sweeping generalization is no longer correct.

On line 100, the authors mention that in the US, a routine one-on-one appointment for cardiac preventive care can take 30 minutes, whereas a cardiologist may see 8-10 patients in a day. However, this example is not applicable in India, neither in the private nor public sector. It would have been more appropriate if the authors had used large public hospitals, for-profit hospitals, and individual doctors to indicate the number of patients they see and the time they devote to each patient. In India, it is uncommon for an outpatient to receive 30 minutes of attention.

Several crucial factors that could have affected the study results are not mentioned by the authors on page 5, line 168. These factors include patients' baseline education level, socioeconomic status, and proximity to the hospital. In addition, information is missing regarding the median hospital registration-consultation time and the median time spent per patient by each consultant in the hospital. It is also unclear how many patients attend the glaucoma clinic daily, the number of consultants and residents available in the outpatient department, and the median waiting time for patients in the hospital OPD.

Page 6, Line 197: Our intervention determined whether groups of patients whose appointments were scheduled proximally were seen separately (in one-on-one appointments) or together (in an SMA). This defeats the purpose of randomization because patients have been preselected.

This sentence is not clear: “but group size in follow-up SMAs varied from 2-6 patients (M=4.262, SD=0.991)” Please spell out “M”- is it mean or median?

Page 8, Line 254: Statistical analysis

Based on prior research [25] and a power analysis that we conducted using data from the pilot study, we proposed an initial target sample size of 1,000 patients (with 500 in each arm), to ensure a study with 90% power (β=0.1) and the ability to detect differences between sample means with 99% confidence (α=0.01). It is not clear how the authors calculated the sample size based on these numbers. We need to have the estimates of the control and the intervention arms. It is also not clear what the phrase “difference between sample means” refers to. What is the mean in the control arm and the intervention arm? Were the data normally distributed? Why not median?

On page 9, line 263, the authors state that linear regression was used to analyze continuous outcomes. However, it is unclear whether the data were normally distributed, which is important in determining whether means or medians should be used to test differences. Additionally, the authors used logistic regression for discrete outcomes, but they did not explain how they computed the odds ratios and 95% confidence intervals or how they performed the analysis.

The authors also mentioned performing subgroup analyses based on patients' sex, age, urban vs. rural residence, education, and comorbidities, but it is unclear whether they collected data on these variables and how the severity of comorbidities was assessed.

Table 1 on page 11 should have presented proportions in absolute numbers instead of decimal points.

The authors reported a difference in patients' overall satisfaction with the appointment, but it is unclear whether this difference is clinically meaningful.

Although the authors stated the study's specific aims, objectives, and hypotheses in the CONSORT checklist of information, they did not mention them specifically in the paper's introduction.

The study design lacks clarity regarding how the computer-generated sets of random allocations were created, who created them, or whether they were sealed in consecutively numbered opaque envelopes. Blinding during intervention and outcome measurement stages was not adequately specified.

The authors did not provide details on the logistic regression model's development, such as which variables were included in the initial model and how the final model was identified and evaluated.

Overall, the study design, analysis, and presentation of results lack quality, with essential elements such as enrollment, randomization, concealment of the randomization sequence, and blinding not adequately addressed. Additionally, the statistical analysis is inadequate, with essential steps for analyzing data and performing linear or logistic regression not correctly followed. Therefore, based on these limitations, the manuscript should be rejected.

6. PLOS authors have the option to publish the peer review history of their article (what does this mean?). If published, this will include your full peer review and any attached files.

**Do you want your identity to be public for this peer review?** For information about this choice, including consent withdrawal, please see our Privacy Policy.

Reviewer #1: No

Reviewer #2: No

---

## [Editor Report · Decision Letter 1]

26 Jun 2023

Evidence from the First Shared Medical Appointments (SMAs) Randomised Controlled Trial in India:  SMAs Increase the Satisfaction, Knowledge, and Medication Compliance of Patients with Glaucoma

PGPH-D-23-00116R1

Dear Prof. Ramdas,

We are pleased to inform you that your manuscript 'Evidence from the First Shared Medical Appointments (SMAs) Randomised Controlled Trial in India:  SMAs Increase the Satisfaction, Knowledge, and Medication Compliance of Patients with Glaucoma' has been provisionally accepted for publication in PLOS Global Public Health.

Best regards,

Madhukar Pai, MD, PhD

Editor-In-Chief
